# Malaria Rapid Diagnostic Tests: Literary Review and Recommendation for a Quality Assurance, Quality Control Algorithm

**DOI:** 10.3390/diagnostics11050768

**Published:** 2021-04-25

**Authors:** Michael J. Kavanaugh, Steven E. Azzam, David M. Rockabrand

**Affiliations:** 1Navy Medicine Leadership and Professional Development Center, 8955 Wood Road, Building 1, Rm 1709, Bethesda, MD 20889, USA; 2Department of Internal Medicine, Walter Reed National Military Medical Center, Bethesda, MD 20814, USA; Steven.e.azzam.mil@mail.mil; 3Department of Biology and Chemistry, Liberty University, 1971 University Blvd, Lynchburg, VA 24515, USA; dmrockabrand@liberty.edu

**Keywords:** malaria rapid diagnostic test (RDT), plasmodium histidine-rich protein (*HRP*), clinical laboratory standards, college of american pathologists, clinical lab improvement act

## Abstract

Malaria rapid diagnostic tests (RDTs) have had an enormous global impact which contributed to the World Health Organization paradigm shift from empiric treatment to obtaining a parasitological diagnosis prior to treatment. Microscopy, the classic standard, requires significant expertise, equipment, electricity, and reagents. Alternatively, RDT’s lower complexity allows utilization in austere environments while achieving similar sensitivities and specificities. Worldwide, there are over 200 different RDT brands that utilize three antigens: Plasmodium histidine-rich protein 2 (*PfHRP-2*), Plasmodium lactate dehydrogenase (pLDH), and Plasmodium aldolase (pALDO). *pfHRP-2* is produced exclusively by *Plasmodium falciparum* and is very *Pf* sensitive, but an alternative antigen or antigen combination is required for regions like Asia with significant *Plasmodium vivax* prevalence. RDT sensitivity also decreases with low parasitemia (<100 parasites/uL), genetic variability, and prozone effect. Thus, proper RDT selection and understanding of test limitations are essential. The Center for Disease Control recommends confirming RDT results by microscopy, but this is challenging, due to the utilization of clinical laboratory standards, like the College of American Pathologists (CAP) and the Clinical Lab Improvement Act (CLIA), and limited recourses. Our focus is to provide quality assurance and quality control strategies for resource-constrained environments and provide education on RDT limitations.

## 1. Introduction

Rapid diagnostic tests (RDTs) have had an enormous impact on the global impact on malaria diagnostics since their emergence in the 1990s [1]. The global burden of malaria includes 229 million cases in 2019 in 87 endemic countries, but the greatest burden remains in Africa, with 215 million cases accounting for 94% of the total cases [2]. Asia accounts for 3% of cases, with India showing huge reductions from 23 million cases in 2000 to 6.3 million cases in 2019. Additionally, Sri Lanka and East Timor have not reported cases since 2015 and 2017, respectively. Worldwide, the overall malaria prevalence has been stable compared to 2018 at 228 million cases [3], but over the last two decades, there has been a significant decrease in mortality from 680,000 in 2000 to 409,000 in 2019 [2].

These improvements are part of a comprehensive elimination and prevention program involving vector control, improved treatments, and better diagnostics [2]. The traditional diagnostic standard of care still remains microscopy which provides the diagnostic capability to include speciation, quantitative assessment of parasitemia, and provides evidence on treatment responsiveness [4]. Thus, the US Center for Disease Control stance on RDTs is that it “does not remove the requirement for microscopy in malaria diagnostics” [5]. However, microscopy is labor-intensive, requires a high degree of skill, requires an expensive diagnostic quality microscope, requires electricity, and routine replenishment of reagents [6]. Thus, microscopy is not always possible in austere environments, and RDTs have dramatically improved the diagnostic capability in resource-limited medical environments [3].

There are over 200 brands of RDTs [7,8], and there were 2.7 billion RDTS sold and 1.9 billion distributed by national malaria programs during the period from 2010–2019 [2,3]. This mass distribution and relative ease of use have resulted in an increase in usage in Africa from 36% to 87% for suspected cases [9]. The magnitude of RDT use has greatly contributed to the 2011 paradigm shift from empiric malaria treatment to obtaining a parasitological diagnosis from either microscopy or RDT prior to treatment [3,6]. This change has been adopted by 44 African countries and has resulted in a huge benefit in decreasing overuse of antimalarials which has a significant cost reduction to a health system, as well as decreasing risk of the development of resistance to artesunate-combination therapies, which currently are still >98% effective [2,6,10,11].

The RDT card is a lateral flow device that uses immunochromatography to detect antigens associated with Malaria [8,12,13]. The card uses a red blood cell lysing agent, and the sample flows via capillary action to identify the antigens by capture antibodies resulting in a positive test line within 15–30 min [12,14]. The *Plasmodium-*specific proteins identified are histidine-rich protein 2 [*HRP2*), *Plasmodium* lactate dehydrogenase (pLDH), and aldolase. The *HRP2* assay utilizes a specific protein that only detects *P. falciparium.* LDH has both a non-specific PpanLDH for all species, as well as more specific. *P. falciparum* (PfLDH) and *P. vivax* (PvLDH) assays. Aldolase is a protein present in all malaria species, and thus, this assay is utilized to diagnose the presence of malaria without speciation [4,14,15]. In general, the *HRP2* based RDTs have reasonable sensitivity for *P. falciparum*, but RDTs, in general, have had variable performance and variable sensitivity. Attributed causes include issues with product design, including difficulty reading the colors on the test bars, operator error, storage issues with the card itself, parasite factors, such as *HRP2* gene deletions, and low parasitemia below the level of detectability and high parasitemia known as the prozone effect [7,13].

## 2. Histidine-Rich Protein (*HRP2*)

*HRP2* is a protein that is produced only by *P. falciparum*, and thus, RDTs that utilize *HRP2* provide the benefit of Pf specificity and with its high sensitivity. Thus, over 80% of all RDTs utilize *HRP2*, and it is a common choice in Africa, with 99.7% of malaria cases being *P. falciparum* [2,14,15]. Infected erythrocytes from *Plasmodium falciparum* produced higher levels of proteins containing histidine than other amino acids, such as methionine or isoleucine [16]. The initial scientific premise for enzyme markers for malaria diagnostics dates back to 1975 with the identification of *Plasmodium* Glutamate Dehydrogenase, which was not present in human erythrocytes [17]. However, better targets, such as *HRP*, now exist. The premise for *HRP* based RDTs is that *Plasmodium falciparum* produces a family of multiple histidine-rich proteins in *HRP2* and *HRP3*. Thus, the development of *HRP*2 monoclonal antibodies to detect the *HRP2* antigen resulted in a cost-effective technology for RDT malaria diagnostics. The first operational RDT cards for *HRP2* were available in the 1990s [1].

The WHO and the Foundation for Innovative New Design (FIND) interactive guide provides a report of RDTs, including false positive and false negative rates, which can be accessed through a spreadsheet at https://www.who.int/malaria/areas/diagnosis/rapid_diagnostic_tests/en/ (Accessed on 11 February 2021) [18,19]. This information assists with the regional selection of RDTs. Although expert microscopy is still considered the gold standard, information on the newer *HRP*-RDTs has often outperformed regional microscopy with sensitivities of 94.1% compared to local hospital microscopic diagnostic capability at 71.8% [6]. Highly skilled tertiary care microscopy was 94% sensitive compared to PCR, which emphasizes that not all microscopy is equivalent despite it being considered the gold standard [6].

At low parasitemia, however (<1000 parasites/uL), the test line is often faint which can be interpreted as a false negative [6,13,14,20]. However, decreased sensitivity with low parasitemia is not unique to *HRP2*. In fact, *HRP2* based RDTs have outperformed pLDH and even regional microsocopy at low parasitemia. For example, in a two-center study in Mozambique and Tanzania, 1898 febrile children were evaluated with 94% sensitivity identified for *HRP2*, but this decreased to 69.9% when parasitemia was <1000 parasites/uL. pLDH was 88% sensitive, but its sensitivity was affected even more profoundly at 45.7% [21]. *HRP2* minimum detection varied from 62–500 parasites/uL, but sensitivity is dramatically higher with increased parasitemia > 500 parasites/uL [13]. However, local laboratory microscopy was only 72–78% sensitive compared to the gold standard of expert microscopy at a reference laboratory [6,21].

## 3. Limitation: Genetic Variation

*HRP2* based RDTs have shown excellent sensitivity in Africa, but variable sensitivity in other regions, with one cause being related to genetic variability causing deletions in the *pHRP2* and *pHRP3* genes [2,7,13]. The *Plasmodium* gene encoding the *HRP2* protein is a single copy subtelomeric gene on chromosome 7 [7]. Although detection is primarily targeting *pHRP2* proteins, *HRP3* cross-reacts with *HRP2*. False negatives occur with gene deletions of *HRP2*. However, there have been identified cases of *HRP2* RDT positive tests with high parasitemia patients despite having molecular evidence of an *HRP2* gene deletion [21]. These patients had an active *HRP*3, and thus, cross-reactivity was triggered, resulting in a positive *HRP2* RDT test at high parasitemia [22]. However, when deletions of both *HRP2* and *HRP3* genes occurred, the test line was not detectable by *HRP2* RDTs, which further supports the cross-reactivity of these proteins on RDTs [14,22].

Mutations resulting in *HRP2* and *HRP3* deletions appear to occur independently of each other [7]. False negatives related to *HRP2* gene deletions were also more prevalent at lower parasitemia [13]. *pHRP2* variability has been seen initially in Asia and Oceana, including Papua New Guinea, Thailand, Philippines, and the Solomon Islands [7,13]. *HRP*2 deletions were also identified in Honduras and Peru with a 40.6% deletion rate identified in Iquitos, Peru, which makes *HRP2* a poor option in that region [13,23]. From 2019–2020, Pf-*HRP*2 and *HRP3* mutations were reported in 15 countries and confirmed in 11 countries, including: China, Equatorial Guinea, Ethiopia, Ghana, Myanmar, Nigeria, Sudan, Uganda, United Kingdom (imported), Tanzania, Zambia [2,24]. As such, the WHO has recommended that countries with *PfHRP2/3* deletions and neighboring countries should “conduct baseline surveys among suspected malaria cases” to determine whether there is a greater than 5% *HRP2* deletion rate resulting in false negatives [2]. Issues of gene deletion can be ameliorated by combination RDTs with *HRP2* combined with either aldolase or pLDH as the second assay, which are not subject to *HRP* gene deletion issues [25]. To date, the WHO has not formally certified any non-*HRP2* combination test as being able to distinguish *P. falciparum* from *P. vivax* [2].

## 4. Limitation: Persistent Positivity and Poor Role as a Test of Treatment

*HRP2* does have an issue with persistent positivity for weeks after effective treatment and resolution of clinical symptoms [20]. Thus, the *HRP2* RDT has limited utility as a test of cure, due to persistent antigenemia. Utilizing a *PfHRP2* assay, 14 day and 21 day false positive rates have been as high as 98.2% and 94.6%, respectively [26,27]. Other studies have evaluated 28 day false positive *HRP2* rates at 26.4%, with effective clearance finally occurring at day 35 [28]. Thus, *HRP2* RDTs are not effective as a test of treatment effectiveness. Additionally, they have limited utility in detecting reinfection within the one month timeframe, due to persistent antigenemia.

## 5. Prozone and Empiric Treatment for Severe Cases

Although RDTs have supported the paradigm shift from empiric treatment to parasitological diagnosis prior to treatment [29], the guidance is still to confirm RDT diagnoses with microscopy [4,5,21]. However, except in the case of reference laboratory level expert microscopy, RDTs often outperform regional microscopy with *HRP2* sensitivities at 94% compared to 72–78% for microscopy [6,21]. WHO Malaria treatment guidelines now recommend against routine presumptive treatment unless strong clinical suspicion of a severe disease without the ability to obtain timely laboratory diagnosis [14,29]. However, it is controversial to withhold treatment with a clinical presentation concerning severe malaria if RDT is negative without microscopy backup. One reason is that RDTs have fewer data in severe malaria compared to studies in uncomplicated malaria [21]. They also do not provide prognostic information or quantitative levels of parasitemia [4,14,21]. The sensitivity for *HRP2* RDTs for various symptoms of severe malaria has ranged from 94–100% with the symptoms of reduced consciousness being on the lower end to hemoglobinuria and jaundice being on the higher end of sensitivity [21]. Furthermore, there have been issues with false negative *HRP2* tests with parasitemia greater than 4% secondary to the prozone effect. Prozone has not been seen with pLDH [14,25]. Dilution studies have shown that the prozone effect has been exhibited in 94% of RDT brands [30]. Thus, there are many benefits to no longer utilizing empiric therapy, but there is still a likely place for presumptive treatment in severe symptomatology.

## 6. Low Parasitemia

Malaria RDT sensitivity decreases with low parasitemia [31]. In one study of 1898 febrile children with microscopy verified samples, *HRP2* and LDH had sensitivities of 94% and 88%, respectively. However, at low parasitemia defined as <1000 parasites/uL, the sensitivity decreased for both assays to 69.9% and 45.7%, respectively [21]. The faintness of the test line has been one recognized challenge with diagnosis [6]. Of note, local microscopists only performed at 78% sensitivity with 84% specificity compared to reference lab expert microscopy [21]. One specific patient population where malaria detection at low parasitemia is clinically very relevant is associated with malaria-related anemia in pregnancy. The concern is that the placenta has a higher parasite load than is detectable in peripheral smears [32]. In one study of 596 Ghanaian pregnant women, local microscopy was only 42% sensitive with *HRP2* RDT 80% sensitive based on malaria PCR as the gold standard [32]. This is an important group to identify because low parasitemia infections may appear asymptomatic to the mother, but have been associated with low birth weights in babies [33]. Although neither the microscopy nor RDT are adequately sensitive, this does lend further evidence that not all microscopy is an equivalent and microscopic backup to RDTs is not quite as the gold standard as the recommendations would imply.

## 7. pLDH

The pLDH assays identify enzymes specific to the malaria glycolytic pathway produced by the *Plasmodium* parasite, and its epitopes are uniquely different from human LDH [10,14]. Benefits of pLDH include the ability to test all types of malaria with PpanLDH or, more specifically, with PfLDH or PvLDH. However, there are no commercially available *P. ovale*, *P. vivax*, or *P. knowlesi* LDH assays [14]. LDH does not have many of the limitations related to gene deletion or prozone, which are seen with *HRP*2 [25]. Furthermore, pLDH is much more effective as a test of cure. The specificity of pLDH is 87% after effective treatment but improves to 92–100% between days 7–42 [19,28]. On account of test of cure, pLDH dramatically outperforms *HRP*2, which has a twoday specificity of 17.3%, seven day of 29.9%, and still only 73.6% specificity at day 28 [28]. PpanLDH has also been able to identify the newest malaria species *P. knowlesi* [34]. However, the sensitivity for *P. knowlesi* is exquisitely affected by parasite counts with a sensitivity of 97% when counts >1000 parasites/uL, but only 25% when parasite count was <1000 parasites/uL [34]. In the same study in Malaysia, pLDH identified *Plasmodium vivax* with a sensitivity of 94% except at low parasitemia (<1000 parasites/uL), sensitivity decreased to 60%, which further emphasizes the major effect of parasite count on pLDH sensitivity [34].

Overall, *Plasmodium falciparum* LDH sensitivities have been variable compared to *HRP*2 at 82.6–88% compared to 93.4–98.5%, respectively [6,25,35]. Other studies have shown better results, such as in 313 patients in Madagascar; the sensitivity was 93% versus 92%, comparing *HRP*2 to pLDH [36]. However, there is a more significant decrease in sensitivity with pLDH at lower parasite density compared to *HRP*2. [14,20]. In regions like the Amazon, *Pf-*LDH may be favored over p*HRP*2 with sensitivities of 98.7% compared to 71.6% associated with p*HRP*2 gene deletions [37]. Additionally, several combination RDT cards utilizing p*HRP*2 with either PpanLDH or Pf/PvLDH have shown higher sensitivity than *HRP*2 alone [15,35].

## 8. Aldolase

Aldolase is an enzyme in the malaria glycolytic pathway that is found in all species of malaria [14,34]. The sensitivity of aldolase-based assays is lower for *P. falciparum* than *HRP*2. Sensitivity of aldolase has been comparable to pLDH [14], but several studies have shown aldolase to be unreliable for *P. vivax*, including 37.5% sensitivity in a combined study of Africa and the Caribbean [38], and 62% in an 84 patient study in Korea [39]. Additionally, an aldolase-based RDT was 23%, 44%, and 56% sensitive for *P. knowlesi, P. falciparum*, and *P. vivax*, respectively, in a 129 patient study in Malaysia [34]. One benefit of aldolase is that the identified genetic mutations for aldolase have not demonstrated any effect on the RDT assay’s sensitivity using the BinaxNOW^®^ assay [39].

## 9. BinaxNOW^TM^

The United States Food and Drug Administration has approved only one RDT (BinaxNow^TM^), which is a combination *HRP*2/Aldolase card. It has reported a sensitivity of 95.3% for *P. falciparum*, and 94.2% specificity [40]. For *Plasmodium vivax;* the sensitivity ranged from 68.9–74.6%, and specificity was 99.8% [40]. In function, the BinaxNOW^TM^ RDT has a *P. falciparum* (T1) line that is linked to *HRP*2, and a pan-malaria T2 line (*Pv, Po*, *or Pm*) that is linked to aldolase. If both T1 and T2 lines are present, it cannot be distinguished by RDT alone whether this is a multi-species infection involving *P. falciparum* plus a non-falciparum species or high *Pf* parasitemia as the aldolase enzyme is preserved in all malaria species, including *P. falciparum.* However, the T2 bar is frequently not present in lower parasite counts as aldolase sensitivity is low with parasite counts <1000 parasites/uL [40]. The United States is very restrictive about utilizing products from other countries that have not met FDA approval; but there are more accurate products with higher sensitivity and specificity than BinaxNOW^TM^. Moreover, RDTs effectiveness has a significant regional variability. Thus, BinaxNOW^TM^ is a reasonable augment when used for travelers returning from Africa, with 99.7% of cases being *P. falciparum* [39].

However, if using a combination *HRP*2/aldolase RDT for Central America, which has 74.1% *P. vivax*, then the *HRP*2′s impact is minimal, and the RDT card is predominantly an aldolase-based card. Detection of the non-falciparum species T2 line is attributed to the aldolase assay. For *Plasmodium malariae*; the sensitivity was 43.8%, and for *Plasmodium ovale,* the sensitivity was 50% [40,41]. Other studies have identified *P*. *vivax* sensitivity as 56% on aldolase-based RDT cards [34]. BinaxNOW^TM^ RDT cards are sensitive to parasite levels, but the *HRP*2 T1 card is more forgiving for low counts than the aldolase T2 card. For *Pf,* the sensitivity decreases from 99.2 to 92.6 and then 89.2 going from >1000 parasites/uL to 500–1000 parasites/uL and 100–500 parasites/uL. For *Pv*, however, its sensitivity is 81% at 1000–5000 parasites/uL, 47.4% at 500–1000 parasites/uL and 23.6% at 100–500 parasites/uL [38]. Thus, the CDC has made the recommendation that “The use of the RDT does not eliminate the need for malaria microscopy”, which was made in part, due to sensitivity of the non-falciparum test and the expectation that a positive malaria case in the US is a travel case and non-endemic [5]. However, in malaria-endemic regions, it is not always possible to follow up the RDT with microscopy, due to resource limitations and skill level of the microscopist. Thus, the BinaxNOW^TM^ has dramatic limitations as a tool by US travelers when febrile in endemic countries, particularly, if used in regions with major non-falciparum prevalence, such as Central America and Asia.

## 10. Evaluation of RDTs and Regional Recommendations

The World Health Organization has developed a Rapid Diagnostic evaluation program that utilizes a spreadsheet under the name Malaria FIND [18]. This interactive program has a database, including all RDTs through September 2018. Data points evaluated include detection rates that are recorded as overall and species-specific. It also lists the false positive rate, which is, again, listed by species. Other items include the heat stability of the RDT card. The WHO has also published guidance for manufacturers on necessary items to test to verify accuracy through the “Guidance on control materials for antigen detecting malaria RDTs [18]. In addition to determining items, such as sensitivity and specificity, there is also a recommendation for evaluating the level of detection to determine how low the parasitemia can be with the continued accuracy of the test [42]. Additionally, some reviews have developed their own scores, such as the performance detection score (PDS) that was a combination of sensitivity and test reproducibility [1].

In evaluating what the most appropriate RDT for a region is, the national ministries of health and local hospitals have to balance RDT performance with cost in developing their malaria programs. For example, a box of 25 BINAXNOW^TM^ tests can cost over $1100, and OmtiMal^®^ can cost approximately $550. Thus, balancing effectiveness with a competitive negotiated rate is necessary. For Africa, an *HRP*2 based card may be appropriate as 99.7% of cases are *Pf*, but the WHO has also made recommendations for countries to evaluate Pf*HRP* 2/3 gene deletion rates and consider alternatives to *HRP*2 RDTs [2]. To date, the following African countries have confirmed Pf*HRP* 2/3 deletions: Ethiopia, Sudan, Uganda, Ghana, Nigeria, and Equatorial Guinea [2], and even countries without verified cases like Tanzania have recommended combination RDTs, including *HRP*/*Pf*LDH [6]. Based on the sensitivity of local laboratory microscopy as opposed to reference expert microscopy, combination RDT may be the most adequate and appropriate in Africa [6,21].

Despite Africa being predominantly *P. falciparum,* there are still cases of non-falciparum malaria that makes combination options better as a pure *HRP*2 strategy will miss these cases [43]. Outside of Africa, single *HRP*2 based RDT regimens alone are not appropriate, and thus, there is a reliance on combination *HRP*/LDH or *HRP*/aldolase. The WHO still only recognizes combinations that include *HRP*2 and does not certify any LDH/aldolase combinations [2]. As both LDH and aldolase have pronounced decrements in sensitivity at low parasitemia; the recommendation for microscopy as a backup in Asia and Central and South America should be performed if possible [6,34,39].

For travelers or medical professionals using the BinaxNOW^TM^ RDT, it would serve as a reasonable diagnostic tool for a traveler to Africa. However, caution should be exercised, due to false negatives that arise in regions with high non-falciparum species, such as the Americas and Asia [37], warranting strategies other than RDT alone [40]. PCR is also certainly a high yield test, particularly in non-falciparum predominant regions, but resource constraints will greatly limit this option [44]. While this is certainly an option for travelers from resource-rich countries, it does not solve the issue of diagnosis of non-falciparum malaria in resource-constrained endemic areas.

When the best product is selected, ministries of health must also ensure that training occurs in all regions as many rural providers have expressed a lack of a comfort level with RDTs. For example, a cross-sectional study in a rural region of Nigeria showed that rural providers preferred empiric treatment and deviated from the national test and treat strategy with a test first pattern of only 7.5%, which further decreased to 3.1% in pregnant women [45]. Alternatively, a study on the Nigeria metropolitan region of Sokoto had shown that 89% of providers were educated on malaria diagnostics, and 80.1% were adhering to the national protocol [46].

While RDT selection is crucial, training and trust in RDTs to implement policy is also important. We reviewed 1076 “Malaria RDT” in a PUBMED Search to develop a table of sensitivity and specificity for the period of 2017–2021. The Malaria FIND initiative already has a comprehensive list of detection rates and false positive rates through September 2018, so this article is attempting to augment and not duplicate their work (Table 1). Articles were excluded from the table if they were not in the time frame or if they were not written in English. Several articles reported positivity rates by RDT as a manner of establishing a prevalence and were not evaluating the RDT itself with a reference arm like expert microscopy or PCR. Thus, the determination of sensitivity and specificity was not performed without a reference standard. There was significant variability in RDT sensitivity especially in studies that were utilizing RDTs for submicroscopic and subclinical infections. Table 2 lists a summary of factors which affect RDT accuracy.

## 11. Quality Assurance/Quality Control (QA/QC) Recommendation

Despite advances in antigen detection and gene sequence amplification technology, microscopic examination of Giemsa-stained blood film remains the gold standard for malaria diagnosis [140]. However, this gold standard of malaria diagnosis only holds when the competency of microscopists and an adequate QA program is guaranteed. Thus, there needs to be an emphasis on external validation of results and training of microscopists [141]. The most accurate and reliable malaria diagnostic results are achieved using Giemsa-stained blood film for microscopy, requiring fresh whole blood samples collected in EDTA anticoagulant blood tubes and must be processed within two hours of collection to limit alteration of red cells and decrease in parasite count [142,143].

With an expert microscopist, malaria microscopy can offer accurate diagnostics with as little as 5–10 parasites/μL, but 50 parasites/μL is a more standard lower limit [144]. In the current age of high-quality light emitting diode (LED) illumination and solar battery chargers, microscopy has become more feasible even in remote areas [145]. However, poor microscopy has long been recognized as a big challenge and is a function of multiple factors, including training and skills maintenance, slide preparation techniques, workload, condition of the microscope, and quality of essential laboratory supplies [146]. Even among laboratories with good infrastructure and training, and among reputed experts, abilities vary significantly.

Therefore, maintaining microscopy as a gold standard requires well-trained, competent microscopists, rigorous maintenance of functional infrastructures, and effective quality assurance/quality control (QA/QC) systems. Training of microscopists and establishing effective QA/QC in malaria diagnosis are key tools for malaria eradication programs. According to Breman (2007), microscopy with a functional QA system was the mainstay of malaria diagnosis during the malaria eradication era in the malaria-eradicated countries [145]. Microscopy is generally sensitive, time-efficient, and can determine parasite-species and quantity; it is also very cost-effective when the initial microscope has been obtained. These features, thus, keep microscopy as the gold standard for the diagnosis of malaria [143].

Training of microscopists and regular competency assessment is critical in malaria diagnosis to ensure the required microscopist skills are not lost over time. External competency assessment and/or retraining for certified competent microscopists is recommended by the WHO at three-year intervals to ensure the accuracy and reliability of malaria microscopy results [143]. This is critical in this era of parasite drug resistance when species determination is of great importance. As multi-drug resistant, *P. falciparum* malaria continues to emerge and as new regimens are developed for differential treatment of *P. falciparum* and other species, accurate species determination becomes critical, and the importance of competency in microscopic diagnosis assumes substantial new weight [147].

A quantitative readout is absolutely required to detect emerging drug resistance, as parasite clearance times lengthen. More so, mRDTs are the most basic tools for parasite-based confirmation of malaria in primary health care settings, and require adequate training and competency in addition to validation against microscopy to ensure the reliability of results. Thus, a national QA/QC program for the training and certification of malaria microscopists is urgently required both for better microscopy and to assure a safe and effective RDT program. Such a program would involve the generation of a large bank of malaria positive stained blood films to use for both initial training, refresher courses, and certification exams.

Substandard malaria RDTs are widespread in resource-limited settings, and lot-to-lot variations may affect the performance of RDTs [148,149,150]. Regulatory approvals from high-income countries are of limited help: For instance, the requirements for the European Union’s conformity label (CE Mark) in the case of malaria RDTs are purely administrative [149]. To overcome this vacuum, WHO and partners organized the ‘Prequalification of Diagnostics Program’: In addition to RDT product dossier assessments, manufacturing sites are inspected for compliance with ISO13485 standards, and an active postmarketing surveillance system has been installed (http://www.who.int/diagnostics_laboratory/evaluations/en/ Accessed on 24 February 2021) [151]. Further, the so-called WHO/FIND Rounds assess RDTs also for diagnostic accuracy (*P. falciparum* and *P. vivax*) and heat stability (http://www.finddiagnostics.org/programs/malaria-afs/malaria/rdt_quality_control/product_testing/ Accessed on 11 February 2021) and WHO/FIND further offer a lot testing program [18,152].

Some countries have a national reference laboratory with services and levels of expertise that exceed the minimum standards. The national laboratory can provide higher levels of microscopy, RDTs, training, reference, quality control/assurance, research and evaluation, standard operating procedures, data management, surveillance, equipment maintenance, and laboratory supervision [153]. In the local laboratory, few tools for QC of individual RDT test kits are available. WHO/FIND produce job aids and appropriate training materials (http://www.finddiagnostics.org/programs/malaria-afs/ Accessed on 24 February 2021) and have developed positive controls (freeze dried recombinant parasite antigen) that are currently under implementation and evaluation [18]. Pending this, there are no controls for RDTs at the bench except for cross-checking with microscopy [154].

The new QA/QC programs should be prioritized and thoroughly evaluated in routine implementation sites to ensure that healthcare workers can identify problems with RDT performance using these tools. In the meantime, periodic supervision and comparison to reference microscopy may be the best currently available option for quality control at the health facility level [155]. The national reference laboratory has a central role in the delivery of diagnostic services at all levels and is responsible for planning, implementation, and monitoring of quality control/assurance. The human and financial resources are seldom available for a national reference laboratory to operate independently of a major hospital or research institute, and should be an essential resource for the national malaria control program [153].

Over the past few years, the Division of Microbiology and Infectious Disease (DMID) of the National Institutes of Health (NIH) in the United States has been working toward improving the performance of clinical research laboratories of institutions conducting NIH-sponsored clinical trials to ensure that results generated from studies will be reliable and acceptable to regulatory bodies. The ultimate goal of the Quality Assurance/Quality Control (QA/QC) activities is to achieve compliance with the College of American Pathologists (CAP) and WHO-AFRO checklists in preparation for accreditation through the implementation of GCLP and the improvement of PT performance [156].

Technology can also play a role in developing good QA/QC activities. The Fionet system uses a device called Deki Reader™, which combines standard mobile devices with custom software to gather demographic patient data, provide guidance to health care workers on conducting testing, taking pictures of completed RDT assays, and transmits data over commercially available cell phone services. The system also contains a web portal for uploading processed RDT images, the transmission of patient demographic information, and remote storage and access of the data. This mobile health technology platform has been successfully used in small programs for quality assurance and quality improvement of malaria diagnosis by community health workers in Kenya [157]. See Table 3 for Strength, Weakness, Opportunity, Threat (SWOT) analysis of QA/QC program. 

## 12. Emerging Diagnostic Technologies

In addition to microscopy, RDTs and PCR, there are several emerging diagnostic technologies that will likely have a role in future comprehensive malaria programs. These technologies include novel photacoustics that utilizes a sensor, which has shown promising results through detection of a specific frequency corresponding to the malaria ring stage [158]. Another novel technology utilizes portable nuclear magnetic resonance (pNMR) technology. NMR has historically been expensive, but there are moves to make the technology smaller and cheaper [159]. Rotating-crystal magneto-optical detection (RMOD) utilizes the different magnetic properties of malaria infected blood because the *Plasmodium* infection results in hemoglobin breakdown that liberates the iron-containing organic crystal called hemozoin [160]. RMOD has shown great promise, including very good levels of detection of *P. vivax* (87% sensitivity and 88% specificity), which could be incorporated into a comprehensive malaria strategy in non-falciparum regions where RDT assays have lower sensitivity. Magnetic resonance relaxometry (MRR) is a tool that uses the relaxation time of protons after magnetic excitement for various diagnostic correlations, including the diagnosis of malaria. Previously, MRR had poor sensitivity at low parasitemia, but cell enrichment techniques have improved its level of detection [161]. Spectroscopy can also be utilized that has promising benefits on its level of detection, but still requires an antigen like LDH and does not necessarily help with the austere challenges [162]. However, there is large desire for portable biosensors that utilize malaria enzymatic assays (*HRP*2, LDH, aldolase), hemozoin, or other malaria biomarkers to provide a readout in a manner analogous to a glucometer [163]. These technologies are working on decreasing cost and size, and accuracy, but RDTs will remain a mainstay for a long time to come.

## 13. Conclusions

The use of RDTs has greatly expanded the ability to diagnose malaria, particularly in resource-limited regions. There are, however, limitations, including variable sensitivity, regional variation secondary to gene deletions, and decreased detection, due to the degree of non-falciparum malaria in a region. *HRP*2 remains the predominant assay in RDTs, and the WHO still only endorses combination RDTs that contain *HRP*2—in part, due to quality, but also related to *P. falciparum* being a clear misdiagnosis. However, regions, such as Central and South America and the Indian subcontinent that have a high *P. vivax*, should consider combinations that also include *P*vLDH/PpanLDH or aldolase. Combinations utilizing *HRP*2/PfLDH/PpanLDH can also be beneficial in Africa, which is predominately *P. falciparum,* but the LDH can increase the sensitivity of *HRP*2 alone as it does not have the same gene deletion or prozone effects as *HRP*2. There are, however, cost and storage considerations to each RDT, and utilization of the WHO Malaria FIND resource is an appropriate way for a health ministry and hospital to select the most appropriate agent. Furthermore, we cannot emphasize enough the importance of developing and implementing a QA/QC program based on high-quality microscopy training and outside verification. RDTs are a great tool for diagnosing and managing malaria, but monitoring its limitations with a QA/QC program and educating clinicians on results can dramatically improve a nation’s malaria program.

## Figures and Tables

**Table 1 diagnostics-11-00768-t001:** 2017–2021 Rapid Diagnostic Test (PUBMED Evaluation). Articles that compared to a reference standard and reported sensitivity and specificity only. This does not include articles that reported an RDT positivity rate without reference.

1st Author	Journal/Year	Brand	Assay Studied	Country	Sensitivity	Specificity
Fagbamigbe AF [47]	Malar J 2019	SD Bioline	*HRP*2	Nigeria	87.6%	75.8%
Enane LA [48]	J Ped Infect Dis 2019	BinaxNOW	*HRP*2/ Aldolase	USA	98.1%	98.8%
Vasquez, AM [49]	PlosOne 2018	SD Bioline	*HRP* Pf/Pv LDH Ppan LDH	Columbia	85.7% 82.8% 77.1%	>99%
Bonko, MDA [50]	Ann Clini Microbiol Antimicrob 2019	Not reported	Pf-*HRP*2	Burkina Faso	72% positivitiy but-no gold standard refernce	59%
Makuuchi, R [51]	BMC Infect Dis 2017	Paracheck	*HRP*2	Malawi	85.7%	80.4%
Odugbemi, B [52]	Inf Dis Poverty 2020	Bioline Pf/Ppan	*HRP*2 pLDH	Mauritania	91%	Not listed
Hawash, Y [44]	Korean J Parasitol 2019	Paramax-3	*HRP*2/ PvLDH aldolase	Saudi Arabia	83.3%	94.2%
McCreesh, P [53]	Malar J 2018	Carestart^™^ Malaria	*HRP*2/ Pf/Ppan LDH	Namibia	85%	99.2%
Naeem, MA [54]	Malar J 2018	SD Bioline	*HRP*2 Pf LDH Ppan LDH	Pakistan	95%	95%
Oyet, C [55]	Malar J 2017	Deki Reader	*HRP*2 Ppan LDH	Tanzania	94.1%	95.6%
Lumbala, C [56]	PloS Negl Trop Dis 2020	SD Bioline	*HRP*2	Uganda DRC	97.3	97.1
Girma, S [57]	Clin Infect Dis 2019	Alere CareStart SD Bioline	*HRP*2 *HRP*2/LDPh *HRP*2	Ethiopia	33.9% # 14.1% 5%	Not reported
Stuck, L [58]	Int J Infect Dis 2020	Not reported	*HRP*2	Tanzania	34%	Not reported
Kumari, P [59]	J Trop Med Hyg 2020	Not reported	Not Reported	India	7.3% #	Not Reported
Okyere B [60]	PloS ONE 2020	Parahit f	*HRP*2	Ghana	100%	100%
Zaw, TZ [61]	Malar J 2017	Carestart	*HRP*2 Ppan LDH	Myanmar	85.7%#	Not reported
Park, SH [62]	Korean J Parasitol 2020	BIOCREDITTM (3 Different subsets)	1. *HRP*/ Ppan LDH 2. Pf/Pv LDH Ppan 3. Pf LDH	India Korea	99% 95.8% 100%	100% 100% 100%
Tambo, M [63]	PLoS ONE 2018	115 different	N/A	Namibia	40.9%	90%
Sitali, L [64]	Malar J 2019	Multiple	N/A	Zambia	75.7%	94.2%
Amoah, LA [65]	BMC Public Health 2019	SD Bioline	*HRP*2	Ghana	54%	89.7%
Costa, MRF [66]	Rev Soc Bras Med Trop 2019	SD-Bioline	*HRP*2 Pf/Pv LDH	Brazil	98.9%	100%
Mbarambara PM [67]	Med Sante Trop 2018	Not listed	Not listed	Congo	97.4%	96.9%
Colborn, J,M [68]	PloS ONE 2020	SD Bioline	*HRP*2	Mozambique	75%	95%
Plucinski, M [69]	Malar J. 2017	Multiple	*HRP*2	Angola Mozambique Haiti	Varied by parasite count	Not recorded
Kiemde, F [70]	Malar J 2017	Not listed	*HRP*2	Burkina Faso	98.2%	58.9%
Plucinski, M [71]	Am J Trop Med Hygeine 2017	SD Bioline	*HRP*2 Pv/Pv LDH	Angola	81%	Not recorded
Rachid-Viana, G,M [72]	PloS ONE 2017	SD Bioline	*HRP*2 Ppan LDH	Peru Bolivia Brazil	95%	Not recorded
Koliopoulos, P [73]	Malar J 2021	Nadal	*HRP*2 Ppan LDH	Tanzania	96.3%	98.1%
Noble, L [74]	BMC Infect Disease 2020	Deki Reader	*HRP*2 Ppan LDH	South Africa	99.8%	97.7%
Li, M [75]	J Infect Dev Ctries 2017	Care Start	*HRP*2 pLDH	Ghana	97.44%	69.52%
Al-Shehri, H. [76]	Malar J 2020	SD Bioline	*HRP*2 PpanLDH	Uganda	94.2%	47.7%
Berzosa, P [24]	Malar J 2020	Nadal	*HRP*2 PpanLDH	Equatorial Guinea	99.7 Pf 95.5 other	99.5%
Mosnier, E [77]	Am J Trop Med Hyg 2020	SD Bioline	*HRP*2 PpanLDH	Brazil, French Guiana	14% #	Not reported
Landier, J [78]	J Clin Microbiol 2018	Alere ultra-sensitive RDT	us*HRP*2	Myanmar	51,4% #	99.4%
Vasquez AM [79]	BMC Pregnancy Childbirth 2020	Alere Ultra-sensitive	hs*HRP*2	Columbia	64.1% #	90%
Grossenbacher, B [80]	Malar J 2020	SD Bioline	*HRP*2 PpanLDH	Tanzania	37% #	99.9%
Maziarz, M [81]	Malar J 2018	Malaria Dual	*HRP*2 PpanLDH	Uganda	92%	100%
Rogier, E [82]	PLoS ONE 2017	First Response	us*HRP*2	Mozambique Angola	86.4% 73.9%	99.52%
Mudare, N [83]	Malar J 2021	Paracheck Pf ICT Malaria Pf	*HRP*2	Zimbabwe	52.4% #	98%
Ajakaye OG [84]	J Parasit Dis 2020	Not listed	Not listed	Nigeria	69.08%	66.67%
Teh RN [85]	Trop Med Health 2019	CareStart	*HRP*2	Cameroon	82.4%	76.6%
Kanwugu ON [86]	J Trop Med 2019	CareStart	*HRP*2/Pf LDH	Ghana	55.6%	93.8%
Nderu D [87]	Parasitol Int 2018	CareStart	Pf*HRP*2/pLDH	Kenya	94%	75%
Jang IK [88]	Am J Trop Med Hyg 2020	Q-Plex	*HRP*2, *Pf* LDH, *Pv* LDH, Pan LDH	Peru	92.7% 71.5%, 46.1%, 83.8%	99.5%
Kiemde F [89]	Malar J 2018	Not listed	*HRP*-2	Burkina Faso	97.5%	52.8%(health facility) 74.2%(lab)
Nkenfou CN [90]	Afr J Infect Dis 2018	SD Bioline	P.f/Pan	Cameroon	75%	48.8%
Mwesigwa J [91]	Malar J 2019	Not listed	HS-RDT	Ghana	38.4%	88.5%
Kiemde F [92]	PLoS ONE 2019	Not listed	Pf*HRP*2 pLDH	Burkina Faso	98.4% 89.3%	74.2% 98.8%
Mfuh KO [93]	Malar J 2019	Not listed	Not listed	Cameroon	78%	94%
Bwire GM [15]	Malar J 2019	CareStart	*HRP*2/pLDH (Pf/pan	Tanzania	99.8%	87.6%
Agarwal R [94]	Cochrane Database Syst Rev 2020	CareStart	Pf/Pv Combo test	Meta-analysis of multiple areas	99%	99%
Eticha T [95]	J Trop Med 2020	CareStart	Pf/Pv Combo test	Ethiopia	97.44%	93.67%,
Wardhani P [96]	Infect Dis Rep 2020	RightSign RDT ScreenPlus	*HRP* II/pLDH *HRP* II/pLDH	Indonesia	100%, 100%	98% 98%
Galatas B [97]	Malar J 2020	SD-Bioline Abbott	*HRP* II/pLDH Pf*HRP*2	Mozambique	61.5 68.2	99.2 99.0
Deutsch-Feldman M [98]	Am J Trop Med Hyg 2018	SD Bioline	*HRP*2	Zambia	45%	Not listed
Abdalla ZA [99]	Trans R Soc Trop Med Hyg 2019	SD Bioline	Ag Pf	Sudan	80.7%	89.3%
Rogier E [100]	J Infect Dis 2020	Unlisted	*HRP*2	Haiti	86.3%	86.3%
Unwin VT [101]	Malar J 2020	CareStart Alere	*HRP*2/pLDH VOM uRDT Pf antigen	Indonesia	22.8% 19.6%	95.5% 98.2%
Gachugia J [102]	Malar J 2020	SD Bioline	P.f/Pan	Kenya	78.1	93.0
Kashosi TM [103]	Pan Afr Med J 2017	SD-Bioline	Pf/Pan	Congo	82.1%	92.0%
Ruas R [104]	Malar J 2017	BinaxNOW	*HRP*-2/Ppan	sub-Saharan Africa	58%	Not listed
Kalinga AK [6]	Malar J 2018	SD Bioline	Pf*HRP*2/pLDH	Tanzania	93.9%	72.0%
Iwuafor AA [105]	Niger Med J 2018	Paracheck	*HRP*-2/Pf	Nigeria	51.4%	73.2%
Niyibizi JB [106]	J Trop Med 2020	CareStart	*HRP*-2	Rwanda	95.0%	59.2%
Mehlotra RK [107]	Am J Trop Med Hyg 2019	SD Bioline	Pf*HRP*2	Madagascar	87%	90%
Natama HM [108]	Sci Rep 2017	SD-Bioline	Pf*HRP*2	Burkina Faso (congenital malaria)	12.5%	99.7%
Coldiron ME [109]	Malar J 2019	SD Bioline CareStart	*HRP*2 pLDH	Nigeria	99% 99%	57.4% 58.0%
Kitutu FE [110]	Malar J 2018	CareStart™	Pf-*HRP*2	Uganda	81.7% (read by drug store) 86.9(read by lab scientist)	90.6, 95.7
Adebisi NA [111]	Pan Afr Med J 2018	CareStart	*HRP*-2	Nigeria	94.6%	91.4%
Leslie T [112]	BMC Med 2017	CareStart	*HRP*2/pLDH	Afghanistan	54.2%	96.8
Ita OI [113]	Trans R Soc Trop Med Hyg 2018	Unlisted	Unlisted	Nigeria	75%	98.80%
Willie N [114]	Am J Trop Med Hyg 2018	SD Bioline	P.f/Pan	Madagascar	87%	90%
Bahk YY [115]	Korean J Parasitol 2018	RapiGEN Malaria Asan EasyTestTM	Pf/Pv pLDH/pLDH *HRP*-2/pLDH	Uganda	87.83% 89.57%	100% 100%
Wogu MN [116]	J Trop Med 2018	CareStart	*HRP*2/pLDH Pf	Nigeria	73.7%	97.3%
Diallo MA [117]	Malar J 2017	CareStart	*HRP*2/pLDH	Senegal	97.3%	94.1%
Bouah-Kamon E [118]	Bull Soc Pathol Exot 2018	SD Bioline	*HRP*2	Côte d’Ivoire	92.7%,	87.1%
Kandie R [119]	BMC Infect Dis 2018	SD Bioline	P.f/Pan	Kenya	91.1%	89.6
Charpentier E [120]	Clin Microbiol Infect 2020	Palutop + 4 Optima	*HRP*-2	Africa	98.3%	99.6
Ba H [121]	Bull Soc Pathol Exot 2017	OptiMal-IT	pLDH	Mauritania	89%	91.1%
Azazy AA [122]	Acta Trop 2018	SD BIOLINE	Pf*HRP*-2/pLDH	Yemen	100.0%	97.3%
Boyce R [123]	Clin Infect Dis 2017	SD BIOLINE	*HRP*2/pLDH	Uganda	97.6%	75.6%
Tegegne B [124]	Malar J 2017	CareStart	*HRP*2/PLDH	Ethiopia	70%	97.4%
Kwenti TE [125]	Infect Dis Poverty 2017	CareStart	*HRP*2/pLDH(Pf/PAN)	Cameroon	88.0%	99.1%
Murungi M [126]	J Clin Microbiol 2017	SD-Bioline	*HRP*2/pLDH(Pf/PAN)	Uganda	99.4%	46.7%
Briand V [127]	Malar J 2020	SD BIOLINE Alere Ultra-sensitive	*HRP*-II/Pf *HRP*2	Benin	44.2% 60.5%	95.7% 93.6%
Feleke DG [128]	BMC Infect Dis 2017	CareStart	*HRP*2/pLDH	Ethiopia	95.4	99.3%
Ruizendaal E [129]	Am J Trop Med Hyg 2017	SD Bioline	*HRP*2/pf	Burkina Faso	81.5%	92.1%
Kanayo II [130]	Afr J Infect Dis 2017	OptiMAL	pLDH	Nigeria	84.2%	95.2%
Ugah UI [131]	Malar J 2017	Carestart, SD Bioline SD Bioline	Not listed Pf PF/PV	Nigeria	25% 25% 68.75	85.29% 94.12% 52.94%
Kozycki CT [132]	Malar J 2017	First Response	pLDH/*HRP*2 *HRP*2	Rwanda	80.2% 89.5%	94.3% 86.2%
Ranadive N [133]	Clin Infect Dis 2017	First Response	*HRP*-2	Swaziland	51.7%-with parasaite density < 100 µL 78.8%-excluding parasite density < 100 µL	94.1% 93.7%
Saha S [134]	Indian J Med Microbiol 2017	SD BIOLINE	P.f/P.v	India	94%	99%
Das S [135]	Am J Trop Med Hyg 2017	SD Bioline Alere	P.f P.f Ultra-Sensitive	Uganda	62% 84%	95% 92%
Adu-Gyasi D [136]	PLoS One 2018	CareStart CareStart SD-Bioline	*HRP*2 *HRP*2/pLDH *HRP*2/pLDH	Ghana	98.2%, 98.2% 98.2%	66.5% 66.5% 69.2%
Quakyi IA [137]	Malar J 2018	First Response SD Bioline	*HRP*2 Pf/Pan-*HRP*II	Ghana.	95.1 96.3	96.6 98.3
Manjurano A [138]	Malar J 2021	SD Bioline SD Bioline	Pf High sensitivity Pf	Tanzania	56.5 69.9	95.0 93.2
Gunasekera, WMKT [139]	Patho Glob Health 2018	CareStart	*HRP*2/pLDH	Sri Lanka	95.95% 100% Pf 92.22 non-Pf	94.92% 97% Pf 99.62% non-Pf

# Study for low prevalence population with asymptomatic, submicroscopic malaria. Diagnosis made with diagnosis by PCR as reference.

**Table 2 diagnostics-11-00768-t002:** Summary of Factors Affecting RDT test accuracy.

**Parasite-Specific Factors**
*HRP2* gene deletion
*HRP3* gene deletion #
Low parasitemia
High parasitemia (prozone effect)
**RDT-Specific Factors**
Assay quality
Heat stability of the RDT card
Age of card or reagent
Lot to lot variability in assay quality
**Operator-Specific Factors**
Operator-Inappropriate placement of reagent or blood drop
Operator-Interpreting faint line
**Miscellaneous Factors**
Regional variation (i.e., *HRP*2 card in a high non-falciparum region)
Prolonged Positivity posttreatment-(most significant with *HRP*2)-poor test of cure and affects the ability to test for reinfection for 4–6 week

# Cross-reactivity with *HRP*3 and *HRP*2 occurs. Despite the assay being directed at *HRP*2; *HRP*3 gene deletions have also been associated with false negative results.

**Table 3 diagnostics-11-00768-t003:** SWOT Analysis of RDT Utilization and QA/QC Program.

**Strength**	**Weaknesses**
Minimal training required Does not require a high level of microscopy training Effective tool in austere environments Rapid results and test run locally Sensitivity and specificity often exceed local hospital microscopists (exception of tertiary level experts) Allowed WHO to recommend moving away from empiric treatment to test and treat strategy.	Variability in quality of RDTs Combination RDTs have higher sensitivity but with higher price *HRP*2 based RDTs are not useful as a test of cure False negatives associated with parasite-specific factors, RDT assay factors, operator and miscellaneous factors (See Table 2) United States limits use to FDA approved devices only—which limits the potential for better products internationally Trust in the result of RDTs in severe cases. Detection in low parasitemia in pregnant women.
**Opportunities**	**Threats**
Opportunity to develop effective RDT QA/QC program utilizing outside verification Development of effective selection tools for local malaria programs using sources like MALARIA FIND Malaria detection has multiple emerging technologies, including malaria biosensors and advances in PCR. Among other utilizations, they can be used as part of a comprehensive program that would still include RDTs at remote sites but with better QA.	Loss of true expertise in the field of microscopy Price of higher quality RDTs may result in purchasing of lower sensitivity products With emerging technologies, the effort to build a strong QA/QC program may lose traction. Competition for research funding with novel diagnostic tools.

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
