# Peer review of "Malaria Rapid Diagnostic Tests: Literary Review and Recommendation for a Quality Assurance, Quality Control Algorithm"

_diagnostics, 2021, doi:10.3390/diagnostics11050768_

Round 1
Reviewer 1 Report
Manuscript ID: Diagnostics-1144596: Malaria Rapid Diagnostic Tests: Literary Review and Recommendation for a Quality Assurance, Quality Control Algorithm: Kavanaugh et. al.
Summary. This manuscript provides an in-depth literature review of malaria RDTs with a critical analysis of antigens used in these RDTs and the factors that might affect their sensitivity. In addition, the review addresses issues related to quality control and quality assurance, which can be very important when generating data about the utility of RDTs in comparison to the gold standard of microscopy. Although the review somewhat relied heavily on the WHO’s Malaria Find Initiative, which has comprehensive data about the RDTs detection rates and false positive rates, it provided further discussion about the evaluation of RDTs with different antigens, how they can be used in different regions with different malaria species, and also gave some insights and recommendations on the RDT’s. So, it will be a useful addition to the literature that is already out there.
Suggested comments
There are a few suggestions for improvement of the manuscript.
- I’d recommend that the authors to please run a checker – there were multiple places with incomplete statements, missing words and punctuations, such as lines 46, 118, 222-223, 422 etc. For line 46 – “may be expensive, requires electricity and routine replenishment of reagents…..” it appears a word was missing, could it have been requires??
- Line 191 – prodome or prozone? – please correct
- I came across statements that I feel need further clarification or discussion – for instance – this statement” In evaluating what is the most appropriate RDT for a region, the national ministries of health and local hospitals will have to balance RDT performance with cost in developing their malaria programs.” Some settings in Africa where budgets are very tight might struggle with securing best performing tests, which might be cost prohibitive. I’m not opposed to the statement, but it needs some context. In addition, although combination RDTs may be the most appropriate in Africa, the issues of cost must also be considered.
- Table 1 is too long and needs to be revised. It’s too excessive in the amount of information and detracts from the main message. If a table cannot fit in 1 page (vertical or horizontal), reconsider it or reduce amount of data. Please make a serious attempt to simplify!
- Table 2 seems unnecessary or can be incorporated in the text. Alternatively, provide frequencies or relative significance of each factor in affecting accuracy of malaria detection. Additionally, you could group or sub-categorized by type, for instance Gene deletion, Operator issues, Quality issues (heat stability, Age or card or reagent).
- This statement could be revised “ Early malariologists, including Ronald Ross, emphasized the importance of external validation of results and training of microscopists “ I think we have advanced enough to know the importance of validation of results. I suggest rephrasing.
Author Response
%MCEPASTEBIN%
Thank you for the recommendations. Please see responses, and appreciate the support in improving this manuscript.
-Line 46-revised with "However, microscopy is labor intensive, requires a high degree of skill, requires an expensive diagnostic quality microscope, requires electricity and routine replenishment of reagents." -118 “These patients had an active HRP3 and thus cross reactivity was triggered the HRP2 RDT test at high parasitemia [22].” Changed to “These patients had an active HRP3 and thus cross reactivity was triggered resulting in a positive HRP2 RDT test at high parasitemia [22].” -223 “It has reported sensitivity of 95.3% for P. falciparum, 94.2% specificity [40].” Added “and” before “94.2% specificity” -422 (now 426) added and “decreased detection due to” the degree of non-falciparum malaria in a region. Also, deleted “with” in the line “HRP2 remains the predominant assay in RDTs “with” and the WHO…” Additional items identified on checking: -70- changed “in RDTs in” to just “RDTs in” -90- Removed “which” from “which provides” to just “provides” -107 changed “increase” to “increased” -133 Added a close bracket “]” -168 changes “RDTs” to “RDT” -178 changed “sensitivity” to “sensitive” -181 changed “or” to “nor” -435 changed “prodrome” to “prozone” -286, added a “,” -385 Excessive spaces before the word and
Brand Assay Studied # Studies Countries Sensitivity Range Specificity Range SD Bioline HRP2 11 Nigeria, Columbia… SD Bioline HRP2/LDH 22 Columbia, Mauritania… Carestart HRP2/LDH The summary really did not provide a value to the reader for the purpose of selecting an RDT for a hospital or national malaria program. Additionally, if Venezuela were to choose SD Bioline, it would want to base a choice based on a regional study like Columbia as opposed to a remote study. If allowed to move to Supplemental material, then that would be our preference and then on line 310, we would need to change (Table 1) to (Supplemental Table 1).
Thank you again for the time you spent reviewing this draft of the manuscript.
%MCEPASTEBIN% %MCEPASTEBIN%
Reviewer 2 Report
This article focus on review RDT in malaria diagnosis. Quality control and recommendation were made accordingly. It is well written, with depth and sufficient references used. There are minor suggestion to improve the manuscript.
- Apart from RDT, we suggest the article to mention and describe (in one dedicated Section) the emergence malaria biosensors (e.g., molecular based, microfluidic, magneto-optical, time-domain NMR). Comment briefly their performance and advantageous, or at least mentioned this in the article. Please use the following articles as guidance,
A Photoacoustic-Surface-Acoustic-Wave Sensor for Ring-Stage Malaria Parasite Detection, S Wang, C Yang, P Preiser… - IEEE Transactions on …, 2020
Micromagnetic resonance relaxometry for rapid label-free malaria diagnosis
WK Peng, et al., Nature medicine 20 (9), 1069
Reply to" Considerations regarding the micromagnetic resonance relaxometry technique for rapid label-free malaria diagnosis"
J Han et al., Nature medicine 21 (12), 1387-1389
A sensitive on‐chip probe–based portable nuclear magnetic resonance for detecting low parasitaemia plasmodium falciparum in human blood, M Gupta, K Singh, DK Lobiyal… - Medical Devices & …, 2020 - Wiley Online Library
Magneto-optical diagnosis of symptomatic malaria in Papua New Guinea, Nature Communications 12, 969 (2021)
Enhancing malaria diagnosis through microfluidic cell enrichment and magnetic resonance relaxometry detection
Tian Fook Kong, et al., Scientific Reports 5, 11425
Recent advances in the development of biosensors for malaria diagnosis, FD Krampa, Y Aniweh, P Kanyong, GA Awandare - Sensors, 2020 - mdpi.com
- What is the number of keywords allowed? Five?
- Typo in abstract (article´s)
- Suggest for the benefit of readers, please summarize in SWOT analysis style using Table 3.
- List of typos – refer to attachment (in highlight yellow)
Author Response
Thank you for the recommendations. Please see responses, and appreciate the support in improving this manuscript.
- In regards to the recommendation on emerging technologies, we added: Section 12. Emerging Diagnostic Technologies In addition to microscopy, RDTs and PCR, there are several emerging diagnostic technologies which will likely have a role in future comprehensive marlaira programs. These technologies include novel photacoustics which utilizes a sensor which has shown promising results through detection of a specific frequency corresponding to the malaria ring stage [158]. Another novel technology utilizes portable nuclear magnetic resonance (pNMR) technology. NMR has historically been expensive, but there are moves to the make the technology smaller and cheaper [159]. Rotating-crystal magneto-optical detection (RMOD) utilizes the different magnetic properties of malaria infected blood because the Plasmodium infection results in hemoglobin breakdown which liberates the iron-containing organic crystal called hemozoin [160]. RMOD has shown great promise including very good levels of detection of P. vivax (87% sensitivity and 88% specificity) which could be incorporated into a comprehensive malaria strategy in non-falciparum regions where RDT assays have lower sensitivity. Magnetic resonance relaxometry (MRR) is a tool which uses the relaxation time of protons after magnetic excitement for varoius diagnostic correlations including the diagnosis of malaria. Previously, MRR had poor sensitivity at low parasitemia, but cell enrichment techniques have improved its level of detection [161]. Spectroscopy can also be utilized which has promising benefits on its level of detection, but still requires an antigen like LDH and does not necessarily help with the austere challenges [162]. However, there is large desire for portable biosensors which utilize malaria enzymatic assays (HRP2, LDH, aldolase), hemozoin or other malaria biomarkers to provide a readout in a manner analogous to a glucometer [163]. These technologies are working on decreasing cost and size and accuracy, but RDTs will remain a mainstay for a long time to come.
- Decreased Key words to 5
- Changed “This article’s focus” to “Our focus”
- Adding SWOT Analysis "Table 3. SWOT Analysis of RDT Utilization and QA/QC Program " See attached word for Table
- Full review of typos
Thank you again for the time you spent reviewing this draft of the manuscript.
